# Submodular Function Minimization with Noisy Evaluation Oracle

**Shinji Ito**[*]

NEC Corporation, The University of Tokyo
i-shinji@nec.com

## Abstract

This paper considers submodular function minimization with *noisy evaluation oracles* that return the function value of a submodular objective with zero-mean additive noise. For this problem, we provide an algorithm that returns an $O(n^{3/2}/\sqrt{T})$-additive approximate solution in expectation, where $n$ and $T$ stand for the size of the problem and the number of oracle calls, respectively. There is no room for reducing this error bound by a factor smaller than $O(1/\sqrt{n})$. Indeed, we show that any algorithm will suffer additive errors of $\Omega(n/\sqrt{T})$ in the worst case. Further, we consider an extended problem setting with *multiple-point feedback* in which we can get the feedback of $k$ function values with each oracle call. Under the additional assumption that each noisy oracle is submodular and that $2 \le k = O(1)$, we provide an algorithm with an $O(n/\sqrt{T})$-additive error bound as well as a worst-case analysis including a lower bound of $\Omega(n/\sqrt{T})$, which together imply that the algorithm achieves an optimal error bound up to a constant.

## 1 Introduction

*Submodular function minimization* (SFM) is an important problem that appears in a wide range of research areas, including image segmentation [31; 33], learning with structured regularization [6], and pricing optimization [26]. The goal in this problem is to find a minimizer of a *submodular function*, a function $f : 2^{[n]} \to \mathbb{R}$ defined on the subsets of a given finite set $[n] := \{1, 2, \ldots, n\}$ and satisfying the following inequality:

$$f(X) + f(Y) \ge f(X \cap Y) + f(X \cup Y). \tag{1}$$

This condition is equivalent to the *diminishing marginal returns* property (see, e.g., [17]): for every $X \subseteq Y \subseteq [n]$ and $i \in [n] \setminus Y$, $f(X \cup \{i\}) - f(X) \ge f(Y \cup \{i\}) - f(Y)$.

Existing studies on SFM assume access to an *evaluation oracle* for $f$ that returns the value $f(X)$ for any $X$ in the feasible region. Under this assumption, a number of efficient algorithms have been discovered, in which the number of oracle calls as well as other computational time is bounded by a polynomial in $n$. The first polynomial-time algorithm was given by Grötschel, Lovász, and Schrijver [19] and was based on the ellipsoid method. Combinatorial strongly polynomial-time algorithms have been independently proposed by Iwata, Fleischer, and Fujishige [28] and by Schrijver [38]. The current best computational time is of $O(n^3 \log^2 n \cdot \text{EO} + n^4 \log^{O(1)} n)$ by Lee et al. [34], where EO denotes the time taken by the evaluation oracle to answer a single query. For approximate optimization, Chakrabarty et al. [11] have proposed an algorithm that finds an $\varepsilon$-additive approximate solution in $\tilde{O}(n^{5/3} \cdot \text{EO}/\varepsilon^2)$ time. The time complexity has been improved to $\tilde{O}(n \cdot \text{EO}/\varepsilon^2)$ by Axelrod et al. [4].

---

[*]This work was supported by JST, ACT-I, Grant Number JPMJPR18U5, Japan.

Table 1: Additive error bounds for submodular minimization with noisy evaluation oracle.

| | Assume $\hat{f}_t$: submodular | Do not assume $\hat{f}_t$: submodular |
|---|---|---|
| single-point feedback | [23][1]: $O\left(\frac{n}{T^{1/3}}\right)$ (if $T = \Omega(n^3)$) <br> [**This paper**]: $O\left(\frac{n^{3/2}}{\sqrt{T}}\right)$ and $\Omega'\left(\frac{n}{\sqrt{T}}\right)$ $\quad(\Omega'(\cdot) := \Omega(\min\{1, \cdot\}))$ | |
| $k$-point feedback <br> $(2 \le k \le n)$ | [**This paper**]: <br> $O\left(\frac{n}{\sqrt{kT}}\right)$ and $\Omega'\left(\frac{n}{\sqrt{2^k T}} + \frac{\sqrt{n}}{\sqrt{T}}\right)$ | [**This paper**]: <br> $O\left(\frac{n^{3/2}}{\sqrt{kT}}\right)$ and $\Omega'\left(\frac{n}{\sqrt{kT}}\right)$ |
| $(n+1)$-point feedback | [23]: <br> $O\left(\frac{\sqrt{n}}{\sqrt{T}}\right)$ and $\Omega'\left(\frac{\sqrt{n}}{\sqrt{T}}\right)$ | [**This paper**]: <br> $O\left(\frac{n}{\sqrt{T}}\right)$ and $\Omega'\left(\frac{\sqrt{n}}{\sqrt{T}}\right)$ |

In some applications, however, evaluation oracles are not always available, and only *noisy* function values are observable. For example, in the pricing optimization problem, let us consider selling $n$ types of products, where the value of the objective function $f(X)$ corresponds to the expected gross profit, and the variable $X \subseteq [n]$ corresponds to the set of discounted products. In this scenario, Ito and Fujimaki [26] have shown that $-f(X)$ is a submodular function under certain assumptions, which means that the problem of maximizing the gross profit $f(X)$ is an example of SFM. In a realistic situation, however, we are not given an explicit form of $f$, and the only thing we can do is to observe the sales of products while changing prices. The observed gross profit does not coincide with its expectation $f(X)$, but changes randomly due to the inherent randomness of purchasing behavior or some temporary events. This means that exact values of $f(X)$ are unavailable, and, consequently, existing works do not directly apply to this situation.

To deal with such problems, we introduce SFM with *noisy evaluation oracles* that return a random value with expectation $f(X)$. In other words, the noisy evaluation oracle $\hat{f}$ returns $\hat{f}(X) = f(X) + \xi$, where $\xi$ is a zero-mean noise that may or may not depend on $X$. We assume access to $T$ independent noisy evaluation oracles $\hat{f}_1, \hat{f}_2, \ldots, \hat{f}_T$ with bounded ranges. We start with the *single-point feedback* setting and then study the more general *multiple-point feedback* (or *k-point feedback*) setting: In the former setting, for each $t \in [T]$, we choose one query $X_t$ to feed $\hat{f}_t$, and get feedback of $\hat{f}_t(X_t)$. In the latter setting, we are given a positive integer $k$, and for each $t$, choose $k$ queries to feed $\hat{f}_t$ and observe $k$ real values of feedback. Such a situation with multiple-point feedback can be assumed in some applications. For example, in the case of pricing optimization for E-commerce, we can get multiple-point feedback by employing the A/B-testing framework, i.e., by showing different prices to randomly divided groups of customers. Note that each $\hat{f}_t$ is not necessarily submodular even if its expectation is submodular.

Our contribution is two-fold, positive results (algorithms, Theorem 1) and negative results (worst-case analyses, Theorem 2): We propose algorithms that return $O(1/\sqrt{T})$-additive approximate solutions, and we show that arbitrary algorithms suffer additive errors of $\Omega(1/\sqrt{T})$ in the worst case. The results are summarized in Table 1 with positive results in $O(\cdot)$ notation and negative ones in $\Omega'(\cdot)$ notation.

As shown in Table 1, for the single-point feedback setting, we propose an algorithm that finds an $O(n^{3/2}/\sqrt{T})$-additive approximate solution. Moreover, there is no room for reducing this additive error bound by a smaller factor than $O(1/\sqrt{n})$. Indeed, our Theorem 2 implies that arbitrary algorithms, including those requiring exponential time and space, suffer at least $\Omega(n/\sqrt{T})$ additive errors. For the $k$-point feedback setting, both the lower and the upper bounds are decreased by $1/\sqrt{k}$ factors, without additional assumptions. Under the assumption that each $\hat{f}_t$ is submodular (Assumption 1), however, the situation changes: Our proposed algorithm achieves $O(n/\sqrt{kT})$-additive error, which is $O(1/\sqrt{n})$-times smaller than without Assumption 1. We also show the lower bound of $\Omega(n/\sqrt{2^k T} + \sqrt{n}/\sqrt{kT})$, which implies that, if $k = O(1)$ or $k = \Omega(n)$, then our algorithm is *optimal* up to constant factors, i.e., no algorithms achieve additive errors of a smaller order.

To construct the algorithms, we combine a convex relaxation technique based on the *Lovász extension* and *stochastic gradient descent* (SGD) method. The Lovász extension for a submodular function is a convex function of which minimizers lead to solutions for SFM. Thanks to this, we can reduce SFM to a convex optimization problem. In this study, we seek a minimizer of the Lovàsz extension by means of SGD, in which we need to construct unbiased estimators of subgradients. The performance of the SGD depends strongly on the variance of subgradient estimators. We present ways for constructing subgradient estimators, and it turns out that Assumption 1 enables us to obtain estimators with smaller variances. The combination of Lovász extension and SGD has been already introduced in the work on *bandit submodular minimization* by Hazan and Kale [23]. Our work, however, considers different problem settings, including multiple-point feedback, and presents tighter and more detailed analyses. Details in the difference are given in Section 2.

A key technique for our lower bounds comes from the proof of regret lower bounds for bandit problems by Auer et al. [3]. Their proof consists of two steps: they first construct a probabilistic distribution of inputs for which it is hard to detect a *good arm* offering a large reward, and then show that any algorithm actually chooses the good arm only infrequently. We follow a line similar to these two steps to prove Theorem 2, in which a number of technical issues arise. In the case of multiple-point feedback, in particular, we need to assess the KL divergence carefully for the observed signals from evaluation oracles.

## 2 Related Work

*Bandit submodular minimization* (BSM) by Hazan and Kale [23] is strongly related to our model. BSM is described as follows: in each iteration $t \in [T]$, a decision maker chooses $X_t \subseteq [n]$ and observe $f_t(X_t)$, where each $f_t : 2^{[n]} \to [-1, 1]$ is a submodular function. In contrast to our model, no stochastic models for $f_t$ are assumed, and the performance of the decision maker is measured by the *regret* defined as $\mathrm{Regret}_T := \sum_{t=1}^T f_t(X_t) - \min_{X \subseteq [n]} \sum_{t=1}^T f_t(X)$. This BSM problem can be regarded as a generalization of our problem with single-point feedback under Assumption 1. Indeed, given a BSM algorithm achieving $\mathrm{Regret}_T \le b(n, T)$ for some function $b$, one can construct an SFM algorithm that returns $b(n, T)/T$-additive approximate solutions (see, e.g., [25]). Since a BSM algorithm with an $O(nT^{2/3})$ regret bound has been proposed in [23], an $O(n/T^{1/3})$-additive approximate algorithm immediately follows, as in Table 1. In BSM, however, it has been left as an open problem whether or not one can achieve $O(n^{O(1)}\sqrt{T})$ regret bounds.

With respect to SFM with an *exact* evaluation oracle, there is a large body of literature [6; 10; 27; 37; 42; 13], in addition to the works mentioned in Section 1. The Fujishige-Wolfe algorithm [17], based on Wolfe's minimum norm point algorithm [42] and the connection between minimum norm points and the SFM shown in [16], is known to have the best empirical performance in many cases [5; 18]. Chakrabarty et al. [10] have shown that the Fujishige-Wolfe algorithm finds an $\varepsilon$-additive approximate solution with a running time of $O(n^2(\mathrm{EO} + n)/\varepsilon^2)$. The same runtime bound can be achieved by a gradient descent approach presented by Bach [6].

For submodular function *maximization* with noisy evaluation oracles, there have been many studies. Hassani et al. [21] provided a nearly 1/2-approximate algorithm for monotone submodular maximization. Singla et al. [41] considered a similar problem with applications to crowdsourcing. Karimi et al. [32] considered maximizing weighted coverage functions, a special case of submodular functions, under matroid constraints, and presented an efficient nearly $(1 - 1/e)$-approximate algorithm. Hassidim and Singer [22] provided a nearly $(1 - 1/e)$-approximate algorithm for monotone submodular maximization with cardinality constraints. Mokhtari et al. [36] showed that a stochastic continuous greedy method works well for monotone submodular function maximization subject to a convex body constraint. For *minimization* problems with similar assumptions, in contrast to maximization problems, only a little literature can be found. Blais et al. [8] considered approximate submodular minimization with an *approximate oracle* model, and presented a polynomial-time algorithm with a high-probability error bound. While their model is more general than ours, their algorithm requires more the computational cost and oracle calls than ours, to achieve a similar error bound. Halabi and Jegelka [20] dealt with minimization of *weakly DR-submodular functions*, which is a class of approximately submodular functions, and provided algorithms with reasonable approximation ratios.

*Zero-order* or *derivative-free convex optimization* [2; 29; 39], optimization problems with evaluation oracle for convex objectives without access to gradients, is also related to our model because Lovász

extensions are convex. For general convex objectives, Agarwal et al. [2], Belloni et al. [7] and Bubeck et al. [9] have proposed algorithms that return $\tilde{O}(1/\sqrt{T})$-additive approximate solutions, ignoring factors of polynomials in $\log T$ and $n^{O(1)}$, where $n$ stands for the dimension of the feasible region. Though the error bounds in these results include factors larger than $O(n^3)$, it has been reported [5; 24] that dependence w.r.t. $n$ can be improved under such additional assumptions as the smoothness and the strong convexity of the objectives. These improved results, however, do not apply to our problems because Lovász extensions are neither smooth nor strongly convex. Multiple-point feedback has been considered in zero-order convex optimization, and some algorithms have been reported to achieve optimal performance in such problem settings [1; 15; 40]. In terms of the lower bound on the additive error, Jamieson et al. [29] and Shamir [39] have shown lower bounds of $\Omega(1/\sqrt{T})$ or $\Omega(1/T)$ for various classes of convex objectives, which, however, do not directly apply to our model.

## 3   Problem Setting

Let $n$ be a positive integer, and let $[n] = \{1, 2, \ldots, n\}$ stand for the finite set consisting of positive integers at most $n$. Let $L \subseteq 2^{[n]}$ be a distributive lattice, i.e., we assume that $X, Y \in L$ implies $X \cap Y, X \cup Y \in L$. Let $f : L \to [-1, 1]$ be a submodular function that we aim to minimize. In our problem setting, we are not given access to exact values of $f$, but given *noisy* evaluation oracles $\{\hat{f}_t\}_{t=1}^T$ of $f$, where $\hat{f}_t$ are random functions from $L$ to $[-1, 1]$ that satisfy $\mathbf{E}[\hat{f}_t(X)] = f(X)$ for all $t = 1, 2, \ldots, T$ and $X \in L$. We also assume that $\hat{f}_1, \hat{f}_2, \ldots, \hat{f}_T$ are independent.

Our goal is to construct algorithms for solving the following problem: First, the algorithm is given the decision set $L$ and the number $T$ of available oracle calls. For $t = 1, 2, \ldots, T$, the algorithm chooses $X_t \in L$ and observes $\hat{f}_t(X_t)$. The chosen query $X_t$ can depend on previous observations $\{\hat{f}_j(X_j)\}_{j=1}^{t-1}$. After $T$ rounds of observation, the algorithm outputs $\hat{X} \in L$. We call this problem a *single-point feedback setting*. In an alternative problem setting, a *multi-point* or *k-point feedback setting*, we are given a parameter $k \geq 2$ in addition to $T$ and $L$. In the $k$-point feedback setting, the algorithm can choose $k$ queries $X_t^{(1)}, X_t^{(2)}, \ldots, X_t^{(k)} \in L$, and, after that, it observes the values $\hat{f}_t(X_t^{(1)}), \hat{f}_t(X_t^{(2)}), \ldots, \hat{f}_t(X_t^{(k)})$ from the evaluation oracle in each round $t \in T$. In both settings, the performance of the algorithm is evaluated in terms of the additive error $E_T$ defined as $E_T = f(\hat{X}) - \min_{X \in L} f(X)$.

A part of our results relies on the following assumption. Note that the following is assumed only when it is explicitly mentioned.

**Assumption 1.** *Assume that each $\hat{f}_t : L \to [-1, 1]$ is submodular and that $k \geq 2$.*

## 4   Our Contribution

Our contribution is two-fold: positive results (Theorem 1) and negative results (Theorem 2).

**Theorem 1.** *Suppose $1 \leq k \leq n + 1$. For the problem with $k$-point feedback, there is an algorithm that returns $\hat{X}$ such that*

$$\mathbf{E}[E_T] = \mathbf{E}[f(\hat{X})] - \min_{X \in L} f(X) = O(n^{3/2}/\sqrt{kT}). \tag{2}$$

*If Assumption 1 holds, there is an algorithm that returns $\hat{X}$ such that*

$$\mathbf{E}[E_T] = \mathbf{E}[f(\hat{X})] - \min_{X \in L} f(X) = O(n/\sqrt{kT}). \tag{3}$$

*The expectation is taken w.r.t. the randomness of oracles $\hat{f}_t$ and the algorithm's internal randomness. In both algorithms, the running time is bounded by $O((k\mathrm{EO} + n \log n)T)$ if $L = 2^{[n]}$, where $\mathrm{EO}$ stands for the time taken by an evaluation oracle to answer a single query.*

If we can choose the number $T$ of oracle calls arbitrarily, we are then able to compute $\varepsilon$-additive approximate solution (in expectation) for arbitrary $\varepsilon > 0$, by means of the algorithm with the error bound (2). The computational time for it is of $O(\frac{n^3}{\varepsilon^2}(\mathrm{EO} + \frac{n}{k} \log n))$. Indeed, to find an $\varepsilon$-additive approximate solution, it suffices to set $T$ so that $\varepsilon = \Theta(\frac{n^{3/2}}{\sqrt{kT}})$, which is equivalent to $T = \Theta(\frac{n^3}{k\varepsilon^2})$.

The computational time is then bounded as $O((k\text{EO}+n\log n)T) = O(\frac{n^3}{\varepsilon^2}(\text{EO}+\frac{n}{k}\log n))$. Similarly, if Assumption 1 holds and the algorithm achieving (3) is used, an $\varepsilon$-additive approximate solution can be found in $O(\frac{n^2}{\varepsilon^2}(\text{EO}+\frac{n}{k}\log n))$ time.

The following theorem gives an insight regarding how tight the above error bounds in Theorem 1 are.

**Theorem 2.** *There is a probability distribution of instances for which any algorithm suffers errors of*

$$\mathbf{E}[E_T] = \mathbf{E}[f(\hat{X}) - \min_{X \in L} f(X)] = \Omega'(n/\sqrt{kT}), \tag{4}$$

*where we denote $\Omega'(\cdot) := \Omega(\min\{1, \cdot\})$. In addition, there is a probability distribution of instances satisfying Assumption 1 for which any algorithm suffers errors of*

$$\mathbf{E}[E_T] = \mathbf{E}[f(\hat{X}) - \min_{X \in L} f(X)] = \Omega'(n/\sqrt{2^kT} + \sqrt{n/T}). \tag{5}$$

*The expectation is taken w.r.t. the randomness of the instance $f$ and oracles $\hat{f}_t$, and the algorithm's internal randomness.*

From (4) in this theorem, we can see that at least $\Omega(\frac{n^2}{\varepsilon^2}\text{EO})$ computational time is required to find an $\varepsilon$-additive approximate solution. This can be shown by an argument similar to that after Theorem 1. For the problem with exact evaluation oracles, on the other hand, Chakrabarty et al. [11] have proposed an algorithm running in $\tilde{O}(\frac{n^{5/3}}{\varepsilon^2}\text{EO})$-time. By comparing these two results, we can see that SFM with noisy oracle is essentially harder than SFM with exact oracle.

# 5 Algorithm

## 5.1 Preliminary

**Lovász extension of submodular function**    For a $[0,1]$-valued vector $x = (x_1, \ldots, x_n)^\top \in [0,1]^d$ and a real value $u \in [0,1]$, define $H_x(u) \subseteq [n]$ to be the set of indices $i$ for which $x_i \geq u$, i.e., $H_x(u) = \{i \in [n] \mid x_i \geq u\}$. For a distributive lattice $L$, define a convex hull $\tilde{L} \subseteq [0,1]^n$ of $L$ as follows: $\tilde{L} = \{x \subseteq [0,1]^n \mid H_x(u) \in L \text{ for all } u \in [0,1]\}$. Given a function $f : L \to \mathbb{R}$, we define the Lovász extension $\tilde{f} : \tilde{L} \to \mathbb{R}$ of $f$ as

$$\tilde{f}(x) = \int_0^1 f(H_x(u))\mathrm{d}u. \tag{6}$$

From the definition, we have $\tilde{f}(\chi_X) = f(X)$ for all $X \in L$, i.e., $\tilde{f}$ is an extension of $f$.[2] The following theorem provides a connection between submodular functions and convex functions:

**Theorem 3** ([35]). *A function $f : L \to \mathbb{R}$ is submodular if and only if $\tilde{f}$ is convex. For a submodular function $f : L \to \mathbb{R}$, we have $\min_{X \in L} f(X) = \min_{x \in \tilde{L}} \tilde{f}(x)$*

For a proof of this theorem, see, e.g., [17; 35].

For $x \in [0,1]^n$, let $\sigma : [n] \to [n]$ be a permutation over $[n]$ such that $x_{\sigma(1)} \geq x_{\sigma(2)} \geq \cdots \geq x_{\sigma(n)}$. For any permutation $\sigma$ over $[n]$, define $S_\sigma(i) = \{\sigma(j) \mid j \leq i\}$. The Lovász extension defined by (6) can then be rewritten as

$$\tilde{f}(x) = f([0]) + \sum_{i=1}^{n}(f(S_\sigma(i)) - f(S_\sigma(i-1)))x_{\sigma(i)} \tag{7}$$

$$= f([0])(1 - x_{\sigma(1)}) + \sum_{i=1}^{n-1} f(S_\sigma(i))(x_{\sigma(i)} - x_{\sigma(i+1)}) + f([n])x_{\sigma(n)}. \tag{8}$$

Similar expression can be found, e.g., Lemma 6.19 in the book [17].

**Subgradient of Lovász extension**    From the above two expressions (7) and (8) of the Lovász extension, we obtain two alternative ways to express its subgradient. For a permutation $\sigma$ over $[n]$ and $i \in \{0, 1, \ldots, n\}$, define $\psi_\sigma(i) \in \{-1, 0, 1\}^n$ as

$$\psi_\sigma(0) = -\chi_{\sigma(1)}, \quad \psi_\sigma(n) = \chi_{\sigma(n)}, \quad \psi_\sigma(i) = \chi_{\sigma(i)} - \chi_{\sigma(i+1)} \quad (i = 1, 2, \ldots, n-1). \tag{9}$$

A subgradient of $\tilde{f}$ at $x$ can then be expressed by $g(\sigma_x)$ defined as

$$g(\sigma) := \sum_{i=1}^{n}(f(S_\sigma(i)) - f(S_\sigma(i-1)))\chi_{\sigma(i)} \tag{10}$$

$$= -f([0])\chi_{\sigma(1)} + \sum_{i=1}^{n-1} f(S_\sigma(i))(\chi_{\sigma(i)} - \chi_{\sigma(i+1)}) + f([n])\chi_{\sigma(n)} = \sum_{i=0}^{n} f(S_\sigma(i))\psi_\sigma(i), \tag{11}$$

where (10) and (11) come from (7) and (8), respectively.

## 5.2 Stochastic Gradient Descent Method

Our algorithm is based on the *stochastic gradient descent* method for $\tilde{f} : \tilde{L} \to [0,1]$. To start with, we initialize $x_1 = \frac{1}{2} \cdot \mathbf{1} \in \tilde{L}$. For $t = 1, 2, \ldots, T$, we update $x_t$ by iteratively calling the oracle $\hat{f}_t$ to obtain $x_{t+1}$. In each update, we construct an *unbiased estimator* $\hat{g}_t$ of a subgradient of $\tilde{f}$ at $x_t$ (a more concrete construction will be given later), and $x_{t+1}$ is given by

$$x_{t+1} = P_{\tilde{L}}(x_t - \eta\hat{g}_t), \tag{12}$$

where $P_{\tilde{L}} : \mathbb{R}^n \to \tilde{L}$ stands for a Euclidean projection to $\tilde{L}$, i.e., $P_{\tilde{L}}(x) \in \arg\min_{y \in \tilde{L}} \|y - x\|_2$, and $\eta > 0$ is a parameter that we can change arbitrarily. We then compute $\bar{x} = \frac{1}{T}\sum_{t=1}^{T} x_t$ and draw $u$ from a uniform distribution over $[0,1]$, and output $\hat{X} = H_{\bar{x}}(u)$. From (6), we have $\mathbf{E}[f(\hat{X})] = \mathbf{E}[\tilde{f}(\bar{x})]$. To analyze the performance of our algorithm, we use the following theorem:

**Theorem 4.** *Let $D \in \mathbb{R}^n$ be a compact convex set containing $0$. For a convex function $\tilde{f} : D \to \mathbb{R}$, let $x_1, \ldots, x_T$ be defined by $x_1 = 0$ and $x_{t+1} = P_D(x_t - \eta\hat{g}_t)$, where $\mathbf{E}[\hat{g}_t|x_t]$ is a subgradient of $\tilde{f}$ at $x_t$ for each $t$. Then, $\bar{x} := \frac{1}{T}\sum_{t=1}^{T} x_t$ satisfies*

$$\mathbf{E}[\tilde{f}(\bar{x})] - \min_{x^* \in D} \tilde{f}(x^*) \leq \frac{1}{T}\left(\frac{\max_{x \in D}\|x\|_2^2}{2\eta} + \frac{\eta}{2}\sum_{t=1}^{T}\mathbf{E}[\|\hat{g}_t\|_2^2]\right). \tag{13}$$

For completeness, we give a proof of this theorem in Appendix A. A similar analysis can be found in, e.g., Lemma 11 of [23]. When setting $D = \tilde{L} - \frac{1}{2} \cdot \mathbf{1}$, we have $\max_{x \in D}\|x\|_2^2 \leq \frac{n}{4}$. From this, Theorems 3 and 4, if $\hat{g}_t$ is bounded as $\mathbf{E}[\|\hat{g}_t\|_2^2] \leq G^2$ for all $t$, we then have $\mathbf{E}[f(\hat{X})] - \min_{X^* \in L} f(X^*) \leq \frac{1}{T}\left(\frac{n}{8\eta} + \frac{\eta}{2}G^2 T\right)$. The performance of the algorithm here depends on $G$, an upper bound on the expected norm of unbiased estimator $\hat{g}_t$. We evaluate the magnitude of $G$ for specific examples of $\hat{g}_t$, in the following subsection.

## 5.3 Unbiased Estimators of Subgradients

In this subsection, we present two different ways to construct unbiased estimators for a subgradient of $\tilde{f}$ that are based on (11) and (10), respectively. The latter is available for the case of multiple-point feedback, i.e., $k \geq 2$, and produces a smaller error bound under Assumption 1. Without such an assumption, the former gives a better error bound. Given $x_t = (x_{t1}, x_{t2}, \ldots, x_{tn})^\top \in [0,1]^n$, let $\sigma : [n] \to [n]$ be a permutation over $[n]$ for which $x_{t\sigma(1)} \geq x_{t\sigma(2)} \geq \cdots \geq x_{t\sigma(n)}$.

**An estimator based on the expression (11)** Suppose $k \in [n+1]$. Consider choosing queries $\{X_t^{(j)}\}_{j=1}^{k}$ randomly as follows: Choose a subset $I_t = \{i_t^{(j)}\}_{j=1}^{k} \subseteq \{0, 1, \ldots, n\}$ of size $k$, uniformly at random, i.e., $I_t$ follows a uniform distribution over the subset family $\{I \subseteq \{0, 1, \ldots, n\} \mid |I| = k\}$. Then let $X_t^{(j)} = S_\sigma(i_t^{(j)}) = \{\sigma(j) \mid j \leq i_t^{(j)}\}$ and observe $\hat{f}_t(X_t^{(j)})$ for $j \in [k]$. Define $\hat{g}_t$ as

$$\hat{g}_t = \frac{n+1}{k}\sum_{j=1}^{k} \hat{f}_t(X_t^{(j)})\psi_\sigma(i_t^{(j)}) = \frac{n+1}{k}\sum_{i \in I_t} \hat{f}_t(S_\sigma(i))\psi_\sigma(i), \tag{14}$$

where $\psi_\sigma(i)$ is defined in (9). Note that $\hat{g}_t$ relies on $x_t$ since $\sigma$ depends on $x_t$. Then, $\hat{g}_t$ is an unbiased estimator of a subgradient and satisfies $\mathbf{E}[\|\hat{g}_t\|_2^2] = O(n^2/k)$:

**Lemma 1.** *Suppose that $\hat{g}_t$ is given by (14). We then have*

$$\mathbf{E}[\hat{g}_t|x_t] \in \partial\tilde{f}(x_t), \quad \mathbf{E}[\|\hat{g}_t\|_2^2] \leq 2(n+1)(n+k)/k. \tag{15}$$

Proofs of all lemmas in this paper are given in Appendix B.

**Algorithm 1** An algorithm for submoudular function minimization with noisy evaluation oracle

---

**Require:** The size $n \geq 1$ of the problem, the number $T \geq 1$ of oracle calls, the number $k \in [n+1]$ of feedback values per oracle call, and the learning rate $\eta > 0$.

1: Set $x_1 = \frac{1}{2} \cdot \mathbf{1}$.
2: **for** $t = 1, 2, \ldots, T$ **do**
3:      Let $\sigma : [n] \to [n]$ be a permutation corresponding to $x_t$, i.e., $x_{t\sigma(1)} \geq \cdots \geq x_{t\sigma(n)}$.
4:      **if** Assumption 1 holds **then**
5:          Choose a subset $J_t \subseteq [n]$ of size $l = \lfloor k/2 \rfloor$, uniformly at random.
6:          Call the evaluation oracle $\hat{f}_t$ to observe $\hat{f}_t(S_\sigma(i))$ and $\hat{f}_t(S_\sigma(i-1))$ for $i \in J_t$.
7:          Construct an unbiased estimator $\hat{g}_t$ of a subgradient of $\tilde{f}$ at $x_t$, as (16).
8:      **else**
9:          Choose a subset $I_t \subseteq \{0, 1, \ldots, n\}$ of size $k$, uniformly at random.
10:         Call the evaluation oracle $\hat{f}_t$ to observe $\hat{f}_t(S_\sigma(i))$ for $i \in I_t$.
11:         Construct an unbiased estimator $\hat{g}_t$ of a subgradient of $\tilde{f}$ at $x_t$, as (14).
12:      **end if**
13:      Compute $x_{t+1}$ from $x_t$ and $\hat{g}_t$ on the basis of (12).
14: **end for**
15: Set $\bar{x} = \frac{1}{T} \sum_{t=1}^{T} x_t$.
16: Draw $u$ from a uniform distribution over $[0, 1]$, and output $\hat{X} = H_{\bar{x}}(u) = \{i \in [n] \mid \bar{x}_i \geq u\}$.

---

**An estimator based on the expression** (10)    Suppose $2 \leq k \leq n + 1$ holds, and let $l$ denote $l = \lfloor k/2 \rfloor \geq 1$. Consider choosing queries $\{X_t^{(j)}\}_{j=1}^{k}$ randomly as follows: Choose a subset $J_t \subseteq \{1, \ldots, n\}$ of size $l$, uniformly at random. Then, set queries $\{X_t^{(j)}\}_{j=1}^{k}$ so that $\bigcup_{i \in J_t} \{S_\sigma(i), S_\sigma(i-1)\} \subseteq \{X_t^{(j)}\}_{j=1}^{k}$, and observe $\hat{f}_t(S_\sigma(i))$ and $\hat{f}_t(S_\sigma(i-1))$ for $i \in J_t$. Define $\hat{g}_t$ as

$$\hat{g}_t = \frac{n}{l} \sum_{i \in J_t} (\hat{f}_t(S_\sigma(i)) - \hat{f}_t(S_\sigma(i-1)) \chi_{\sigma(i)}. \tag{16}$$

Then, $\hat{g}_t$ is an unbiased estimator of a subgradient and satisfies $\mathbf{E}[\|\hat{g}_t\|_2^2] = O(n^2/k)$, and if $\hat{f}_t$ is a submodular function, then $\mathbf{E}[\|\hat{g}_t\|_2^2] = O(n/k)$ holds.

**Lemma 2.** *Suppose that $\hat{g}_t$ is given by (16). We then have*

$$\mathbf{E}[\hat{g}_t | x_t] \in \partial \tilde{f}(x_t), \quad \mathbf{E}[\|\hat{g}_t\|_2^2] \leq 4n^2/l \leq 12n^2/k. \tag{17}$$

*In addition, if $\hat{f}_t$ is a submodular function, we then have*

$$\mathbf{E}[\|\hat{g}_t\|_2^2] \leq 16n/l \leq 48n/k. \tag{18}$$

A key factor in the advantage of the estimator defined by (16) is that the vector $(f_t(S_\sigma(i)) - f_t(S_\sigma(i-1)))_{i=1}^{n} \in \mathbb{R}^n$ has a smaller norm than $(f_t(S_\sigma(i)))_{i=0}^{n} \in \mathbb{R}^{n+1}$, which is implied by Lemma 8 in [23] or Lemma 1 in [30].

## 5.4 Proof of Theorem 1

By combining SGD described in Section 5.2 and unbiased estimators defined by (14) or (16), we obtain Algorithm 1. Let us evaluate the additive errors for this algorithm. Note that we have $\mathbf{E}[\tilde{f}(\bar{x})] - \min_{x* \in \tilde{L}} \tilde{f}(x^*) = \mathbf{E}[f(\hat{X})] - \min_{X^* \in L} f(X^*)$ from (6) and Theorem 3.

Suppose $\hat{X}$ is produced by Algorithm 1 in which Steps 9–11 are chosen. From Theorem 4 and Lemma 1, we have $\mathbf{E}[f(\hat{X})] - \min_{X^* \in L} f(X^*) \leq \frac{1}{T} \left( \frac{n}{8\eta} + \frac{\eta T (n+1)(n+k)}{k} \right)$. The right-hand side is minimized when $\eta$ is chosen as $\eta = \sqrt{\frac{kn}{8T(n+1)(n+k)}}$. We then have $\mathbf{E}[f(\hat{X})] - \min_{X^* \in L} f(X^*) \leq \sqrt{\frac{n(n+1)(n+k)}{2kT}} = O(\frac{n^{3/2}}{kT})$, which proves (2).

Suppose that Assumption 1 holds and that $\hat{X}$ is produced by Algorithm 1, where Steps 5–7 are chosen. From Theorem 4 and Lemma 1, we have $\mathbf{E}[f(\hat{X})] - \min_{X^* \in L} f(X^*) \leq \frac{1}{T} \left( \frac{n}{8\eta} + \frac{24\eta T n}{k} \right)$.

The right-hand side is minimized when $\eta$ is chosen as $\eta = \sqrt{\frac{k}{192T}}$. We then have $\mathbf{E}[f(\hat{X})] - \min_{X^* \in L} f(X^*) \leq \frac{\sqrt{12}n}{\sqrt{kT}} = O(\frac{n}{\sqrt{kT}})$, which proves (3).

The computational time of Algorithm 1 can be evaluated as follows: Step 3 can be conducted by a sorting algorithm, which takes $O(n \log n)$ time. Step 5 can be done by generating uniform random numbers over $[m]$ for $m = n, n - 1, \ldots, n - k + 1$, which takes $O(k \log n)$ times. Step 6 requires $O(k\text{EO})$ time computation. Step 7 can be computed with $O(n)$ arithmetic operations. Steps 9–11 are similar to Steps 5–7. Step 13 takes $O(n)$ time since $x_t - \eta \hat{g}_t$ can be computed with $O(n)$ arithmetic operations and since $P_{\tilde{L}}(x)$ has an explicit form. Hence, Steps 2–14 require $O((n \log n + k \log n + k\text{EO} + n + n) \cdot T) = O((k\text{EO} + n \log n)T)$ time. The other steps do not require time greater than this. Therefore, the overall time complexity is of $O((k\text{EO} + n \log n)T)$.

## 6 Lower Bound

### 6.1 Construction of Hard Instance

Define $h_i : 2^{[n]} \to \{-1, 1\}$ as $h_i(X) := \begin{cases} -1 & (i \in X) \\ 1 & (i \notin X) \end{cases}$ . for $i \in [n]$. Fix a subset $S^* \subseteq [n]$ and a positive real value $\varepsilon \in [0, 1]$. Consider the following procedure that produces a function $\hat{f} : 2^{[n]} \to \{-1, 1\}$: (1) Choose $i \in [n]$ uniformly at random, and set $s = 1$ with probability $\frac{1-\varepsilon}{2}$, $s = -1$ with probability $\frac{1+\varepsilon}{2}$. (2) Define $\hat{f} : 2^{[n]} \to \{-1, 1\}$ by $\hat{f}(X) = s \cdot h_i(S^* \triangle X) = s \cdot h_i(S^*)h_i(X)$, where $S^* \triangle X$ stands for the symmetric difference between $S^*$ and $X$, i.e., $S^* \triangle X = (S^* \setminus X) \cup (X \setminus S^*)$. Let $F(S^*, \varepsilon)$ denote the distribution of functions generated by the above procedure. A similar construction can be found in [14], which is for a lower bound of bandit linear optimization.

In addition, define $F'(S^*, \varepsilon)$ similarly, so that all function values of $f \sim F'(S^*, \varepsilon)$ are *stochastically independent*: Choose $i_X \in [n]$ and $s_X$ with the probability defined as the above, independently for all $X \subseteq [n]$, and define $\hat{f}(X) = s_X \cdot h_{i_X}(S^*)h_i(X)$. Let $F'(S^*, \varepsilon)$ denote the distribution of functions generated by this procedure. Note that each $\hat{f}$ generated from $F(S^*, \varepsilon)$ is a modular function and that this does not always hold for $F'(S^*, \varepsilon)$. If $D_{S^*} = F(S^*, \varepsilon)$ or if $D_{S^*} = F'(S^*, \varepsilon)$, the expectation of $\hat{f} \sim D_{S^*}$ is a submodular function expressed as

$$f_{S^*, \varepsilon}(X) := \mathop{\mathbf{E}}_{\hat{f} \sim D_{S^*}}[\hat{f}(X)] = -\frac{\varepsilon}{n} \sum_{i=1}^n h_i(S^*)h_i(X) = \frac{\varepsilon}{n}(2|S^* \triangle X| - n), \qquad (19)$$

where the second equality comes from $\mathbf{E}[s] = \mathbf{E}[s_X] = \frac{1-\varepsilon}{2} - \frac{1+\varepsilon}{2} = -\varepsilon$.

### 6.2 Proof of Theorem 2

To prove Theorem 2, we start with bounding the additive error from below by means of KL divergences. Fix $X^{(1)}, X^{(2)}, \ldots, X^{(k)} \subseteq [n]$ arbitrarily. For a class $\{D_{S^*} \mid S^* \subseteq [n]\}$ of distributions over $\{\hat{f} : 2^{[n]} \to \{-1, 1\}\}$, let $P_{S^*}$ denote the distribution of $y(\hat{f}) = (\hat{f}(X^{(1)}), \hat{f}(X^{(2)}) \ldots, \hat{f}(X^{(k)}))^\top \in \mathbb{R}^k$ for $\hat{f} \sim D_{S^*}$. We then have the following:

**Lemma 3.** *Suppose that a class of distributions $\{D_{S^*} \mid S^* \subseteq [d]\}$ satisfies (19) for all $S^* \subseteq [d]$. In addition, suppose that the following holds for arbitrary $S^*, X^{(1)}, X^{(2)}, \ldots, X^{(k)} \subseteq [n]$:*

$$\sum_{i=1}^n D_{\text{KL}}(P_{S^*} || P_{S^* \triangle \{i\}}) \leq \frac{n}{2T}. \qquad (20)$$

*If $S^*$ is chosen uniformly at random from $2^{[n]}$, and $\hat{f}_t$ follows $D_{S^*}$ i.i.d. for $t = 1, 2, \ldots, T$, then any algorithm suffers an additive error of $\mathbf{E}[E_T] = \mathbf{E}\left[f_{S^*, \varepsilon}(\hat{X}) - \min_{S \in 2^{[n]}} f_{S^*, \varepsilon}(S)\right] \geq \frac{\varepsilon}{2}$, where the expectation is taken w.r.t. $S^*$, $\hat{f}_t$, and the internal randomness of algorithms.*

Intuitively, the condition (20) means that the distribution of the observed values $y$ does not change much even if the optimal solution $S^*$ is perturbed. Consequently, under the condition (20), it is hard for any algorithm to detect $S^*$. Sufficient conditions for (20) are given in the following two lemmas:

**Lemma 4.** *Suppose that $\{P_{S^*}\}$ is defined by $D_{S^*} = F'(S^*, \varepsilon)$ for $0 \leq \varepsilon \leq \min\{\frac{1}{6}, \frac{n}{\sqrt{8kT}}\}$. Then (20) holds for arbitrary $S^*, X^{(1)}, \ldots, X^{(k)} \subseteq [n]$.*

**Lemma 5.** *Suppose that $\{P_{S^*}\}$ is defined by $D_{S^*} = F(S^*, \varepsilon)$ for $0 \leq \varepsilon \leq \min\{\frac{1}{6}, n\sqrt{\frac{5}{24T\min\{2^k, 2n\}}}\}$. Then* (20) *holds for arbitrary* $S^*, X^{(1)}, \ldots, X^{(k)} \subseteq [n]$.

Theorem 2 can be proven by combining Lemmas 3, 4, and 5. From Lemmas 3 and 4, if $S^*$ is chosen uniformly at random from $2^{[n]}$ and if $\hat{f}_t$ follows $F'(S^*, \varepsilon)$ with $\varepsilon = \min\{\frac{1}{6}, \frac{n}{\sqrt{8kT}}\}$, i.i.d. for $t \in [T]$, we then have $\mathbf{E}[E_T] \geq \frac{\varepsilon}{2} = \min\{\frac{1}{12}, \frac{n}{\sqrt{32kT}}\} = \Omega'(\frac{n}{\sqrt{kT}})$, which proves (4). If $k \geq 2$ and if $S^*$ is chosen uniformly at random from $2^{[n]}$, and $\hat{f}_t$ follows $F(S^*, \varepsilon)$ with $\varepsilon = \min\{\frac{1}{6}, n\sqrt{\frac{5}{24T\min\{2^k, 2n\}}}\}$, i.i.d. for $t \in [T]$, then Assumption 1 is satisfied since $\hat{f}_t \sim F(S^*, \varepsilon)$ is a submodular. Further, from Lemmas 3 and 5, we have $\mathbf{E}[E_T] \geq \frac{\varepsilon}{2} = \min\{\frac{1}{12}, n\sqrt{\frac{5}{96T\min\{2^k, 2n\}}}\} = \Omega'(\max\{\frac{n}{\sqrt{T2^k}}, \sqrt{\frac{n}{T}}\}) = \Omega'(\frac{n}{\sqrt{T2^k}} + \sqrt{\frac{n}{T}})$, which proves (5).

## 7 Conclusion and Open Questions

We have introduced submodular function minimization with noisy evaluation oracle, and have provided algorithms and lower bounds, which together implies that the proposed algorithms achieve nearly optimal additive errors, modulo $O(\sqrt{n})$ factors. For the special cases of $k$-point feedback settings, in which $2 \leq k = O(1)$ and each noisy evaluation oracle itself is a submodular function, we have provided a tight error bound. For the other cases, we leave it as an open question to find tight bounds.

## Acknowledgment

The author thanks Satoru Iwata for valuable discussions and for pointing out important literature. The author also thanks reviewers for many helpful comments and suggestions.

## Footnotes

[1] This work applies to more general problem settings than ours, *bandit submodular minimization* and *online submodular minimization*. See Section 2 for details.

[2] $\chi_X \in \{0,1\}^n$ denotes the indicator vector of $X$, i.e., $(\chi_X)_i = 1$ for $i \in X$ and $(\chi_X)_i = 0$ for $i \in [n] \setminus X$.

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
