[Supplementary Material]

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

# A  Proof of Theorem 4

Since $\tilde{f}$ is convex, from Jensen's inequality, we have

$$\tilde{f}(\bar{x}) = \tilde{f}\left(\frac{1}{T}\sum_{t=1}^{T} x_t\right) \le \frac{1}{T}\sum_{t=1}^{T}\tilde{f}(x_t). \tag{21}$$

Denote $g_t = \mathbf{E}[\hat{g}_t|x_t]$. since $g_t$ is a subgradient of the convex function $\tilde{f}$ at $x_t$, we have

$$\tilde{f}(x_t) - \tilde{f}(x^*) \le g_t^\top (x_t - x^*), \tag{22}$$

for all $t \in [T]$ and $x^* \in D$. By combining (21) and (22), we obtain

$$\mathbf{E}[\tilde{f}(\bar{x})] - \tilde{f}(x^*) \le \frac{1}{T}\mathbf{E}\left[\sum_{t=1}^{T} g_t^\top (x_t - x^*)\right] \le \frac{1}{T}\mathbf{E}\left[\sum_{t=1}^{T} \hat{g}_t^\top (x_t - x^*)\right]. \tag{23}$$

Since we have $\|x_t - \eta\hat{g}_t - x^*\|_2^2 = \|x_t - x^*\|_2^2 - 2\eta\hat{g}_t^\top (x_t - x^*) + \eta^2\|\hat{g}_t\|_2^2$, the value of $\hat{g}_t^\top (x_t - x^*)$ can be bounded as follows:

$$\hat{g}_t^\top (x_t - x^*) = \frac{1}{2\eta}(\|x_t - x^*\|_2^2 - \|x_t - \eta\hat{g}_t - x^*\|_2^2) + \frac{\eta}{2}\|\hat{g}_t\|_2^2. \tag{24}$$

From the Pythagorean theorem (see, e.g., Theorem 2.1 in [25]), since $x^* \in D$, we have $\|x_t - \eta\hat{g}_t - x^*\|_2 \ge \|P_D(x_t - \eta\hat{g}_t) - x^*\|_2 = \|x_{t+1} - x^*\|_2$. From this and (24), we have

$$\hat{g}_t^\top (x_t - x^*) \le \frac{1}{2\eta}(\|x_t - x^*\|_2^2 - \|x_{t+1} - x^*\|_2^2) + \frac{\eta}{2}\|\hat{g}_t\|_2^2.$$

By taking summation of this for $t = 1, 2, \ldots, T$, we obtain

$$\sum_{t=1}^{T} \hat{g}_t^\top (x_t - x^*) \le \frac{1}{2\eta}\sum_{t=1}^{T}(\|x_t - x^*\|_2^2 - \|x_{t+1} - x^*\|_2^2) + \frac{\eta}{2}\sum_{t=1}^{T}\|\hat{g}_t\|_2^2$$

$$= \frac{1}{2\eta}(\|x_1 - x^*\|_2^2 - \|x_{T+1} - x^*\|_2^2) + \frac{\eta}{2}\sum_{t=1}^{T}\|\hat{g}_t\|_2^2$$

$$\le \frac{1}{2\eta}\max_{x \in D}\|x\|_2^2 + \frac{\eta}{2}\sum_{t=1}^{T}\|\hat{g}_t\|_2^2,$$

where the last inequality follows from $x_1 = 0$, $x^* \in D$ and $\|x_{T+1} - x^*\|_2^2 \ge 0$. From this and (23), we have

$$\mathbf{E}[\tilde{f}(\bar{x})] - \tilde{f}(x^*) \le \frac{1}{T}\left(\frac{1}{2\eta}\max_{x \in D}\|x\|_2^2 + \frac{\eta}{2}\sum_{t=1}^{T}\mathbf{E}[\|\hat{g}_t\|_2^2]\right).$$

Since this holds for arbitrary $x^* \in D$, we have (13). $\qquad\square$

# B  Proof of Lemmas

## B.1  Proof of Lemma 1

*Proof.* From the definition (14) of $\hat{g}_t$, its expectation may be expressed as

$$\mathbf{E}[\hat{g}_t|x_t] = \frac{n+1}{k}\mathbf{E}\left[\sum_{i \in I_t}\hat{f}_t(S_\sigma(i))\psi_\sigma(i)\right] = \frac{n+1}{k}\sum_{i=0}^{n}\mathrm{Prob}[i \in I_t]\mathbf{E}\left[\hat{f}_t(S_\sigma(i))\right]\psi_\sigma(i)$$

$$= \frac{n+1}{k}\sum_{i=0}^{n}\mathrm{Prob}[i \in I_t]f_t(S_\sigma(i))\psi_\sigma(i), \tag{25}$$

where the last equality comes from the assumption of $\mathbf{E}[\hat{f}_t(X)] = f(X)$. Since $I_t$ is chosen uniformly at random from all subsets of $\{0, 1, \ldots, n\}$ having size $k$, for each $i \in \{0, 1, \ldots, n\}$, the

probability that $i \in I_t$ is $\binom{n}{k-1}/\binom{n+1}{k} = \frac{k}{n+1}$. Substituting this into (25), we obtain $\mathbf{E}[\hat{g}_t|x_t] = \sum_{i=0}^{n} f_t(S_\sigma(i))\psi_\sigma(i)$. This vector is equal to $g(\sigma)$ given in (11), which is a subgradient of $\tilde{f}$ at $x_t$. We next evaluate the expectation of $\|\hat{g}_t\|_2^2$. From the definition (14) of $\hat{g}_t$, we have

$$\mathbf{E}[\|\hat{g}_t\|_2^2] = \frac{(n+1)^2}{k^2} \mathbf{E}\left[\sum_{i \in I_t}\sum_{j \in I_t} \hat{f}_t(S_\sigma(i))\hat{f}_t(S_\sigma(j))\psi_\sigma(i)^\top \psi_\sigma(j)\right]$$

$$= \frac{(n+1)^2}{k^2}\sum_{i=0}^{n}\sum_{j=0}^{n}\mathrm{Prob}[i,j \in I_t]\mathbf{E}\left[\hat{f}_t(S_\sigma(i))\hat{f}_t(S_\sigma(j))\right]\psi_\sigma(i)^\top \psi_\sigma(j)$$

$$\leq \frac{(n+1)^2}{k^2}\sum_{i=0}^{n}\sum_{j=0}^{n}|\mathrm{Prob}[i,j \in I_t]\psi_\sigma(i)^\top \psi_\sigma(j)|, \tag{26}$$

where the last inequality follows from the assumption that $\hat{f}_t(X) \in [-1,1]$. From the definition of $I_t$, we have

$$\mathrm{Prob}[i,j \in I_t] = \begin{cases} \binom{n}{k-1}/\binom{n+1}{k} = \frac{k}{n+1} & (i = j) \\ \binom{n-1}{k-2}/\binom{n+1}{k} = \frac{k(k-1)}{n(n+1)} & (i \neq j) \end{cases}. \tag{27}$$

Further, from the definition (9) of $\psi_\sigma(i)$, we have

$$\psi_\sigma(i)^\top \psi_\sigma(j) = \begin{cases} 2 & (i = j) \\ -1 & (|i-j| = 1) \\ 0 & (|i-j| \geq 2) \end{cases}. \tag{28}$$

Combining (26), (27) and (28), we have

$$\mathbf{E}[\|\hat{g}_t\|_2^2] \leq \frac{(n+1)^2}{k^2}\left(\sum_{i=0}^{n}\mathrm{Prob}[i \in I_t]\cdot 2 + \sum_{i=0}^{n-1}\mathrm{Prob}[i,i+1 \in I_t] + \sum_{i=1}^{n}\mathrm{Prob}[i,i-1 \in I_t]\right)$$

$$= \frac{(n+1)^2}{k^2}\left(\sum_{i=0}^{n}\frac{2k}{n+1} + \sum_{i=0}^{n-1}\frac{k(k-1)}{n(n+1)} + \sum_{i=1}^{n}\frac{k(k-1)}{n(n+1)}\right)$$

$$= \frac{(n+1)^2}{k^2}\left(2k + \frac{2k(k-1)}{n+1}\right) = \frac{2(n+1)(n+k)}{k},$$

which proves (15). $\qquad\square$

## B.2 Proof of Lemma 2

*Proof.* From the definition (16) of $\hat{g}_t$, its expectation may be expressed as

$$\mathbf{E}[\hat{g}_t|x_t] = \frac{n}{l}\mathbf{E}\left[\sum_{i \in J_t}(\hat{f}_t(S_\sigma(i)) - \hat{f}_t(S_\sigma(i-1))\chi_{\sigma(i)}\right]$$

$$= \frac{n}{l}\sum_{i=1}^{n}\mathrm{Prob}[i \in J_t]\mathbf{E}\left[\hat{f}_t(S_\sigma(i)) - \hat{f}_t(S_\sigma(i-1)\right]\chi_{\sigma(i)}$$

$$= \sum_{i=1}^{n}(f_t(S_\sigma(i)) - f_t(S_\sigma(i-1))\chi_{\sigma(i)},$$

where the last equality follows from the fact that $\mathrm{Prob}[i \in J_t] = \binom{n-1}{l-1}/\binom{n}{l} = \frac{l}{n}$ holds for all $i \in [n]$ and the assumption of $\mathbf{E}[\hat{f}_t(X)] = f(X)$. This vector is equal to the $g(\sigma)$ given in (10), which is a subgradient of $\tilde{f}$ at $x_t$. We next evaluate the expectation of $\|\hat{g}_t\|_2^2$. From the definition (16) of $\hat{g}_t$, we

have

$$\mathbf{E}[\|\hat{g}_t\|_2^2] = \frac{n^2}{l^2} \mathbf{E}\left[\sum_{i \in J_t} \sum_{j \in J_t} (\hat{f}_t(S_\sigma(i)) - \hat{f}_t(S_\sigma(i-1)))(\hat{f}_t(S_\sigma(j)) - \hat{f}_t(S_\sigma(j-1)))\chi_{\sigma(i)}^\top \chi_{\sigma(j)}\right]$$

$$= \frac{n^2}{l^2} \sum_{i=1}^{n} \operatorname{Prob}[i \in J_t] \, \mathbf{E}\left[(\hat{f}_t(S_\sigma(i)) - \hat{f}_t(S_\sigma(i-1)))^2\right]$$

$$= \frac{n}{l} \sum_{i=1}^{n} \mathbf{E}\left[(\hat{f}_t(S_\sigma(i)) - \hat{f}_t(S_\sigma(i-1)))^2\right] \le \frac{4n^2}{l}, \tag{29}$$

where the second equality comes from that $\chi_i^\top \chi_j = \begin{cases} 1 & (i = j) \\ 0 & (i \ne j) \end{cases}$, the third equality comes from that $\operatorname{Prob}[i \in J_t] = \frac{l}{n}$, and the inequality follows from the assumption that $\hat{f}_t(X) \in [-1, 1]$. From this and the fact that $l = \lfloor k/2 \rfloor \ge k/3$, (17) follows. If $\hat{f}_t$ is a submodular function, from Lemma 8 in [23], or Lemma 1 in [30], we have $\sum_{i=1}^{n}(\hat{f}_t(S_\sigma(i)) - \hat{f}_t(S_\sigma(i-1)))^2 \le 16$. From this and (29), we obtain (18). $\qquad\square$

### B.3    Proof of Lemma 3

*Proof.* Since any randomized algorithms can be regarded as a convex combination of deterministic algorithms, it suffices to consider only deterministic algorithms. Fix a deterministic algorithm and let $\{(X_t^{(1)}, \ldots, X_t^{(k)})\}_{t=1}^{T}$ denote the queries generated by it. Denote $y_t = (\hat{f}_t(X_t^{(1)}), \ldots, \hat{f}_t(X_t^{(k)}))^\top \in \{-1, 1\}^k$, the input to the algorithm. From (19), we have

$$f_{S^*,\varepsilon}(\hat{X}) = -\frac{\varepsilon}{n} \sum_{i=1}^{n} h_i(S^* \triangle \hat{X}). \tag{30}$$

To evaluate the above value, we fix $i \in [n]$, and focus on $\displaystyle\mathbf{E}_{S^*}\left[\mathbf{E}_{f_t \sim D_{S^*}}[h_i(S^* \triangle \hat{X})]\right]$. Since $S^* \triangle \{i\}$ follows a uniform distribution over $2^{[n]}$, the same distribution as of $S^*$, we have

$$\mathbf{E}_{S^*}\left[\mathbf{E}_{f_t \sim D_{S^*}}[h_i(S^* \triangle \hat{X})]\right] = \mathbf{E}_{S^*}\left[\mathbf{E}_{f_t \sim D_{S^* \triangle \{i\}}}[h_i((S^* \triangle \{i\}) \triangle \hat{X})]\right] = -\mathbf{E}_{S^*}\left[\mathbf{E}_{f_t \sim D_{S^* \triangle \{i\}}}[h_i(S^* \triangle \hat{X})]\right]$$

where the second equality comes from the definition of $h_i$. Hence, we have

$$\mathbf{E}_{S^*}\left[\mathbf{E}_{f_t \sim D_{S^*}}[h_i(S^* \triangle \hat{X})]\right] = \frac{1}{2}\mathbf{E}_{S^*}\left[\mathbf{E}_{f_t \sim D_{S^*}}[h_i(S^* \triangle \hat{X})] - \mathbf{E}_{f_t \sim D_{S^* \triangle \{i\}}}[h_i(S^* \triangle \hat{X})]\right]$$

$$= \frac{1}{2}\mathbf{E}_{S^*}\left[h_i(S^*)\left(\mathbf{E}_{f_t \sim D_{S^*}}[h_i(\hat{X})] - \mathbf{E}_{f_t \sim D_{S^* \triangle \{i\}}}[h_i(\hat{X})]\right)\right], \tag{31}$$

where the second equality comes from $h_i(S \triangle S') = h_i(S)h_i(S')$. Since $\hat{X}$ is determined by $Y^T = (y_1, \ldots, y_T) \in \{-1, 1\}^{k \times T}$, there is a function $\phi_i : \{-1, 1\}^{k \times T} \to \{-1, 1\}$ such that $h_i(\hat{X}) = \phi_i(Y^T)$. Let $Q$ and $Q_i'$ denote the probability distributions of $Y^T$ for $f_t \sim F(S^*, \varepsilon)$ and

$f_t \sim D(S^* \triangle \{i\}, \varepsilon)$, respectively. We then have

$$\left| \mathop{\mathbf{E}}_{f_t \sim D_{S^*}} [h_i(\hat{X})] - \mathop{\mathbf{E}}_{f_t \sim D_{S^* \triangle \{i\}}} [h_i(\hat{X})] \right|$$

$$= \left| \sum_{y \in \{-1,1\}^T} Q(y)\phi_i(y) - \sum_{y \in \{-1,1\}^T} Q'_i(y)\phi_i(y) \right|$$

$$= \left| \sum_{y \in \{-1,1\}^T} (Q(y) - Q'_i(y))\phi_i(y) \right|$$

$$\leq \sum_{y \in \{-1,1\}^T} |Q(y) - Q'_i(y)| = \|Q - Q'_i\|_1 \tag{32}$$

where $\|Q - Q'_i\|_1$ stands for the total variation distance between $Q$ and $Q'_i$. From Pinsker's inequality (see, e.g., Theorem 12.6.1 in [12]), the total variation distance can be bounded by means of the KL divergence as

$$\|Q - Q'_i\|_1 \leq \sqrt{2D_{\mathrm{KL}}(Q\|Q'_i)}. \tag{33}$$

Combining equations (30) – (33) and $|h_i(S^*)| = 1$, we have

$$- \mathop{\mathbf{E}}_{S^*, f_t \sim D_{S^*}} [f_{S^*, \varepsilon}(\hat{X})] \leq \frac{\varepsilon}{2n} \mathop{\mathbf{E}}_{S^*} \left[ \sum_{i=1}^{n} \sqrt{2D_{\mathrm{KL}}(Q\|Q'_i)} \right] \leq \frac{\varepsilon}{2} \mathop{\mathbf{E}}_{S^*} \left[ \sqrt{\frac{2}{n} \sum_{i=1}^{n} D_{\mathrm{KL}}(Q\|Q'_i)} \right], \tag{34}$$

where the second inequality follows from the Jensen's inequality and the concavity of $\sqrt{x}$. Denote $Y^t = (y_1, \ldots, y_t)$. From the chain rule of KL divergence (see, e.g., Theorem 2.5.3 in [12]), we have

$$\sum_{i=1}^{n} D_{\mathrm{KL}}(Q\|Q'_i) = \sum_{t=1}^{T} \mathop{\mathbf{E}}_{Y^{t-1} \sim Q} \left[ \sum_{i=1}^{n} \mathop{D_{\mathrm{KL}}}_{y_t \sim Q|_{Y^{t-1}}, \, y'_t \sim Q'_i|_{Y^{t-1}}} (y_t\|y'_t) \right]. \tag{35}$$

When $Y^{t-1}$ is fixed, $X_t^{(1)}, \ldots, X_t^{(k)}$ are also fixed since we have fixed a deterministic algorithm. Hence, from the assumption of (20), we have

$$\sum_{i=1}^{n} \mathop{D_{\mathrm{KL}}}_{y_t \sim Q|_{Y^{t-1}}, \, y'_t \sim Q'_i|_{Y^{t-1}}} (y_t\|y'_t) \leq \frac{n}{2T}.$$

By combining this, (34), and (35), we have

$$- \mathop{\mathbf{E}}_{S^*, f_t \sim D_{S^*}} [f_{S^*, \varepsilon}(\hat{X})] \leq \frac{\varepsilon}{2}.$$

We also here have

$$\min_{S \in 2^{[n]}} f_{S^*, \varepsilon}(S) = f_{S^*, \varepsilon}(S^*) = -\varepsilon.$$

From the above two inequalities, the expected additive error is bounded as

$$\mathbf{E} \left[ f_{S^*, \varepsilon}(\hat{X}) - \min_{S \in 2^{[n]}} f_{S^*, \varepsilon}(S) \right] \geq -\frac{\varepsilon}{2} + \varepsilon = \frac{\varepsilon}{2},$$

which accomplishes the proof. $\qquad\square$

## B.4 Proof of Lemma 4

*Proof.* Suppose that $X^{(1)}, \ldots, X^{(k)}$ are distinct. Then, since $\hat{f}(X^{(j)})$ with $\hat{f} \sim F'(S^*, \varepsilon)$ are stochastically independent for $j = 1, \ldots, k$, we have

$$D_{\mathrm{KL}}(P_{S^*}\|P_{S^* \triangle \{i\}}) = \sum_{j=1}^{k} \mathop{D_{\mathrm{KL}}}_{\hat{f} \sim F'(S^*, \varepsilon), \hat{f}' \sim F'(S^* \triangle \{i\}, \varepsilon)} (\hat{f}(X^{(j)})\|\hat{f}'(X^{(j)})). \tag{36}$$

From the definition of $F'(S^*, \varepsilon)$, $\hat{f}(X^{(j)})$ follows Bernoulli distributions of parameters $\theta := \frac{1}{2} + \frac{\varepsilon}{2n}(2|S^* \triangle X_t| - n)$ and $\theta' := \frac{1}{2} + \frac{\varepsilon}{2n}(2|(S^* \triangle \{i\}) \triangle X_t| - n)$ for $\hat{f} \sim F(S^*, \varepsilon)$ and $\hat{f} \sim F(S^* \triangle \{i\}, \varepsilon)$, respectively. Since $\theta, \theta' \in [\frac{1}{3}, \frac{2}{3}]$ and $|\theta - \theta'| = \frac{\varepsilon}{n} \leq \frac{1}{6}$, we have

$$
\begin{aligned}
\underset{\hat{f} \sim F'(S^*, \varepsilon), \hat{f}' \sim F'(S^* \triangle \{i\}, \varepsilon)}{D_{\mathrm{KL}}} (\hat{f}(X^{(j)}) \| \hat{f}'(X^{(j)})) &= -\theta \log \frac{\theta'}{\theta} - (1 - \theta) \log \frac{1 - \theta'}{1 - \theta} \\
&\leq -\theta \left( \frac{\theta' - \theta}{\theta} - \frac{4}{5} \left( \frac{\theta' - \theta}{\theta} \right)^2 \right) - (1 - \theta) \left( \frac{\theta - \theta'}{1 - \theta} - \frac{4}{5} \left( \frac{\theta - \theta'}{1 - \theta} \right)^2 \right) \\
&= \frac{4}{5}(\theta' - \theta)^2 \left( \frac{1}{\theta} + \frac{1}{1 - \theta} \right) \leq \frac{4\varepsilon^2}{n^2} \leq \frac{1}{2kT},
\end{aligned}
\tag{37}
$$

where the first inequality follows from $-\log(1 + x) \leq -x + \frac{4}{5}x^2$ for $|x| \leq \frac{1}{2}$ and the last inequality follows from the assumption of $\varepsilon \leq \frac{n}{\sqrt{8kT}}$. By combining (36) and (37), we obtain $D_{\mathrm{KL}}(P_{S^*} \| P_{S^* \triangle \{i\}}) \leq \frac{1}{2T}$ for all $i \in [n]$, which implies that (20) holds. For the case that $X^{(1)}, \dots, X^{(k)}$ are not distinct, i.e., when $\{X^{(1)}, \dots, X^{(k)}\}$ consists of $k' < k$ elements, we can show $D_{\mathrm{KL}}(P_{S^*} \| P_{S^* \triangle \{i\}}) \leq \frac{k'}{2kT}$ in the same way, from which (20) follows. $\qquad \square$

## B.5   Proof of Lemma 5

*Proof.* From the definition of $F(S^*, \varepsilon)$, $\hat{f} \sim F(S^*, \varepsilon)$ can be expressed as $\hat{f}(X) = V \cdot h_I(S^* \triangle X)$, where $I$ and $V$ follows distributions over $[n]$ and $\{-1, 1\}$, respectively. Hence, if $S^*$ and $X^{(1)}, \dots, X^{(k)}$ are fixed, $y(\hat{f}) := (\hat{f}(X^{(1)}), \dots, \hat{f}(X^{(k)})) \in \{-1, 1\}^k$ changes depending only on $I$ and $V$. Therefore, there exists a function $\lambda : [n] \times \{-1, 1\} \to \{-1, 1\}^k$ such that $y(\hat{f}) = \lambda(I, V)$. Denote $Z = \mathrm{range}(\lambda) = \{\lambda(i, s) \in \{-1, 1\}^k \mid i \in [n], s \in \{-1, 1\}\}$. Then we have

$$
|Z| \leq \min\{|\{-1, 1\}^k|, |[n] \times \{-1, 1\}|\} = \min\{2^k, 2n\}.
\tag{38}
$$

From the definition of $F'(S^*, \varepsilon)$, for any fixed $z \in Z$ and $i \in [n]$, we have

$$
\left| \underset{\hat{f} \sim F(S^*, \varepsilon)}{\mathrm{Prob}} [y(\hat{f}) = z] - \underset{\hat{f} \sim F(S^* \triangle \{i\}, \varepsilon)}{\mathrm{Prob}} [y(\hat{f}) = z] \right| = \begin{cases} \frac{\varepsilon}{n} & (z \in \{\lambda(i, -1), \lambda(i, 1)\}) \\ 0 & (z \notin \{\lambda(i, -1), \lambda(i, 1)\}) \end{cases}.
\tag{39}
$$

From this and the definition of the KL divergence, we have

$$
\begin{aligned}
D_{\mathrm{KL}}(P_{S^*} \| P_{S^* \triangle \{i\}}) &= -\sum_{z \in Z} \underset{\hat{f} \sim F(S^*, \varepsilon)}{\mathrm{Prob}} [y(\hat{f}) = z] \log \frac{\underset{\hat{f} \sim F(S^* \triangle \{i\}, \varepsilon)}{\mathrm{Prob}} [y(\hat{f}) = z]}{\underset{\hat{f} \sim F(S^*, \varepsilon)}{\mathrm{Prob}} [y(\hat{f}) = z]} \\
&\leq \frac{4}{5} \sum_{z \in Z} \frac{\left( \underset{\hat{f} \sim F(S^*, \varepsilon)}{\mathrm{Prob}} [y_t = y] - \underset{\hat{f} \sim F(S^* \triangle \{i\}, \varepsilon)}{\mathrm{Prob}} [y(\hat{f}) = z] \right)^2}{\underset{\hat{f} \sim F(S^*, \varepsilon)}{\mathrm{Prob}} [y(\hat{f}) = z]} \\
&= \frac{4\varepsilon^2}{5n^2} \left( \frac{1}{\underset{\hat{f} \sim F(S^*, \varepsilon)}{\mathrm{Prob}} [y(\hat{f}) = \lambda(i, -1)]} + \frac{1}{\underset{\hat{f} \sim F(S^*, \varepsilon)}{\mathrm{Prob}} [y(\hat{f}) = \lambda(i, 1)]} \right),
\end{aligned}
$$

where the inequality follows from a calculation used in (37), and the last equality follows from (39). By taking a sum of the above for $i \in [n]$, we obtain the following:

$$\sum_{i=1}^{n} D_{\text{KL}}(P_{S^*} || P_{S^* \triangle \{i\}}) \leq \frac{4\varepsilon^2}{5n^2} \sum_{i=1}^{n} \left( \frac{1}{\underset{\hat{f} \sim F(S^*, \varepsilon)}{\text{Prob}} [y(\hat{f}) = \lambda(i, -1)]} + \frac{1}{\underset{\hat{f} \sim F(S^*, \varepsilon)}{\text{Prob}} [y(\hat{f}) = \lambda(i, 1)]} \right)$$

$$= \frac{4\varepsilon^2}{5n^2} \sum_{z \in Z} \left( \frac{|\{i \in [n] \mid \lambda(i, 1) = z\}|}{\underset{\hat{f} \sim F(S^*, \varepsilon)}{\text{Prob}} [y(\hat{f}) = z]} + \frac{|\{i \in [n] \mid \lambda(i, -1) = z\}|}{\underset{\hat{f} \sim F(S^*, \varepsilon)}{\text{Prob}} [y(\hat{f}) = z]} \right)$$

$$= \frac{4\varepsilon^2}{5n^2} \sum_{z \in Z} \frac{|\{(i, s) \in [n] \times \{-1, 1\} \mid \lambda(i, s) = z\}|}{\underset{\hat{f} \sim F(S^*, \varepsilon)}{\text{Prob}} [y(\hat{f}) = z]}.$$

Since $I$ follows a uniform distribution over $[n]$ and $V$ follows a Bernoulli distribution with the parameter $\frac{1-\varepsilon}{2}$, we have $\text{Prob}[I = i, V = s] \geq \frac{1-\varepsilon}{2n} \geq \frac{1}{3n}$ for all $i \in [n]$ and $s \in \{-1, 1\}$. Hence, we have

$$\underset{\hat{f} \sim F(S^*, \varepsilon)}{\text{Prob}} [y(\hat{f}) = z] = \underset{\hat{f} \sim F(S^*, \varepsilon)}{\text{Prob}} [\lambda(I, V) = z] \geq \frac{|\{(i, s) \in [n] \times \{-1, 1\} \mid \lambda(i, s) = z\}|}{3n}.$$

Combining the above two equations, (38), and the assumption of $\varepsilon \leq n\sqrt{\frac{5}{24T \min\{2^k, 2n\}}}$, we obtain

$$\sum_{i=1}^{n} \underset{y_t \sim P|_{Y^{t-1}}, \, y'_t \sim P'_i|_{Y^{t-1}}}{D_{\text{KL}}} (y_t || y'_t) \leq \frac{4\varepsilon^2}{5n^2} \sum_{y \in Z} 3n = \frac{12\varepsilon^2 |Z|}{5n} \leq \frac{n}{2T}.$$

$\square$