[Reviews · NeurIPS 2019]

Reviewer 1



The paper is written well. Overall, the problem considered in this paper is important for today's potentially large-scale applications where instead of the actual function value (that is given as an expectation or as a finite-sum over the whole data) we need to resort to stochastic estimates (i.e. an iid sample from the distribution or randomly chosen mini-batches). While the problem formulation is novel, the main novelty is in providing the unbiased estimates of the gradient of the lovasz extension and the rest follows immediately due to the convexity of the extension. The lower bound is also novel. A note regarding related work: The concept of submodular optimization with noisy (unbiased) oracles has been introduced before in two papers: (i) Gradient methods for submodular optimization, (ii) Stochastic Submodular Maximization: The Case of Coverage. To the best of my knowledge these works use SGD albeit in the context of maximization. Also, the work "Submodular Optimization under Noise" considers noisy (worst-case noise) oracles and should be cited as related work.

Reviewer 2



On line 178, the authors should acknowledge this construction of subgradient. This paper [1] provided this construction several years ago. However, I am not sure if it is the earliest paper that designed this construction. The authors should try to search the book [2]. [1] Djolonga, Josip, and Andreas Krause. "From MAP to marginals: Variational inference in bayesian submodular models." Advances in Neural Information Processing Systems. 2014. [2] Fujishige, Satoru. Submodular functions and optimization. Vol. 58. Elsevier, 2005. Using Lovasz extension and stochastic projected gradient descent are standard. However, the authors should acknowledge previous works for this method that solves the minimization problem of submodular functions. The major contribution of this paper is the two estimators presented in Sec 5.3. The two estimators look so natural that it is surprising to me that nobody would not have considered this before. If there are previous works that only use a gradient estimator, the authors should have a detailed literature review of these works. On line 208, I was confused why I_t is a subset of {0, 1, ..., n}. The ground set is {1, 2, ..., n}. Why could I_t contain 0? Furthermore, it is extremely confusing that the construction from line 207 to line 210 seems not to rely on x_t. On lines 217-218, the notation \bigcup_{i\in J_t} {S_\sigma(i), S_\sigma(i-1)} is perplexing. S_\sigma(i) and S_\sigma(i-1) are two sets and therefore {S_\sigma(i), S_\sigma(i-1)} is a set of two sets. I could not decipher what this sentence tries to convey. J_t seems to be like I_t basically, also picked uniformly at random. It is just that its size l is large (=\lceil k/2\rceil). What is the high-level intuition that the estimator (16) is better? The authors should explain the high-level idea carefully. I checked the proofs of Lemma 1 and Lemma 2 regarding the expectation and variance of the proposed gradient estimators. On line 375 (Appendix B.1), the authors should explain that 1{i\in I_t} and \hat{f}_t(S_\sigma(i)) are independent. It seems to me that the high-level idea that (16) works better is the inequality on line 395. I have a couple of suggestions regarding it. First, the lemma (Lemma 8 in [17], or Lemma 1 in [25]), even including its proof if short, should be included for completeness. Second, the authors should state clearly how the inequality helps get a smaller variance and why multi-point evaluations are crucial here. Furthermore, the authors should justify in the paper that the assumption regarding multi-point evaluations is a realistic assumption. The authors may want to do it by giving examples where one does have access to random submodular functions (the evaluation of k points is from the same randomly sampled submodular function). The presentation of Sec 6.1 should be significantly improved. The authors present a distribution in the first paragraph, which is F(S^*, \epsilon). They present another distribution F'(S^*, \epsilon). What did the authors mean by "stochastically independent"? Is it the usual independence? Why pick i_X and s_X? Are they used to make sure that evaluation of \hat{f} is independent given different X? Why is this construction (sampled from F'(S^*, \epsilon)) a submodular function? If it is submodular, is it indeed a modular function? What is the connection between \hat{f} on line 248 and \hat{f} on line 254? There are many technical details missing so that I could not evaluate the correctness of this paper and it is hard for me to make a favorable recommendation of this paper before the authors present their detailed explanations and arguments. It would be great if the authors could clarify all these aforementioned points in their response and incorporate them into the paper.

Reviewer 3



Originality: The model is new. Though it's somehow similar to the bandit model, the new model captures exploits a salient feature that often appear in practical applications of submodular function minimization that is not captured by the bandit model. Quality: The proofs seem to be correct. The proofs are not very hard technically as we already have a sheer amount of proof techniques developed in online convex optimization. Clarity: It was easy to follow the argument. Significance: Although the proofs are not very hard, I think the model is worth studying. Line 172: The range of f is missing

[Author Response · NeurIPS 2019]

Dear Reviewer #1:

> The concept of submodular optimization with noisy (unbiased) oracles has been introduced before

Thanks for pointing out related works. As the reviewer mentioned, for submodular function *maximization* problems, there have been works considering noisy oracles. We will mention three papers you suggested in the revised version.

> Closing the gap between lower and upper bound.

Yes, this is a significant future work. This seems to be a difficult problem as it has been open to shave off polynomial gaps between lower and upper bounds in zeroth order convex optimization problems with noisy evaluation as well.

> Also, experiments showing that SGD with ... considerably faster in large-scale settings would be beneficial.

We agree with this suggestion. We are leaving empirical evaluation as an important future task.

Dear Reviewer #2:

Thanks for providing many comments and questions that help improve our paper. We will modify the manuscript by incorporating the following points:

> On line 178, the authors should acknowledge this construction of subgradient.

We will add the references you suggested in the revised version. The current manuscript mentions no specific literature because we consider that these expressions of subgradient are a sort of folklore and because the oldest literature is not clear. We can find a similar expression in Lemma 6.19 of the book by Fujishige.

> Using Lovasz extension and stochastic projected gradient descent are standard.

To our knowledge, the only literature that considers the combination of Lovasz extension and the *stochastic* projected gradient is [17], which is mentioned in lines 74-76, 86-96. Combining Lovasz extension and (exact) gradient methods can be found in more literature such as [4] and [5]. We will add a more detailed review in the revised version.

> On line 208, I was confused why $I_t$ is a subset of 0, 1, ..., n.;    This should be $\{1, 2, \ldots, n\}$. We will fix it.

> Furthermore, it is extremely confusing that the construction from line 207 to line 210 seems not to rely on $x_t$.

Because $\sigma$ depends on $x_t$, $\hat{g}_t$ relies on $x_t$. We emphasize this in the revised version.

> On lines 217-218, the notation $\bigcup_{i \in J_t} \{S_\sigma(i), S_\sigma(i-1)\}$ is perplexing.;    This means that the set of queries $\{X_t^{(j)}\}_{j=1}^k$ must include $S_\sigma(i)$ and $S_\sigma(i-1)$ for all $i \in J_t$, i.e., $\hat{f}_t(S_\sigma(i))$ and $\hat{f}_t(S_\sigma(i-1))$ must be evaluated for all $i \in J_t$.

> What is the high-level intuition that the estimator (16) is better?

As the reviewer pointed out, a key factor is that the vector $(f_t(S_\sigma(i)) - f_t(S_\sigma(i-1)))_{i=1}^n$ has a smaller norm than $(f_t(S_\sigma(i)))_{i=0}^n$, which is implied by Lemma 8 in [17] or Lemma 1 in [25]. We will modify the manuscript as suggested.

> the authors should justify in the paper that the assumption regarding multi-point evaluations is a realistic assumption.

For example, in the case of pricing optimization (lines 33-42) for E-commerce, we can get multiple-point feedback by employing the A/B-testing framework, i.e., by showing different prices to randomly divided groups of customers.

> What did the authors mean by "stochastically independent"? ;    We mean usual independence.

> Why pick $i_X$ and $s_X$? Are they used to make sure that evaluation of $\hat{f}$ is independent given different X?

Yes, we pick $i_X$ and $s_X$ for each $X$ to let $\hat{f}(X)$ be independent for different $X$. This is needed in the proof of Lemma 4, to obtain a larger regret lower bound. (In line 254, $h_{i_X}(S^*)h_i(X)$ should be replaced with $h_{i_X}(S^*)h_{i_X}(X)$.)

> Why is this construction (sampled from $F'(S^*, \varepsilon)$) a submodular function? ... is it indeed a modular function?

The expectation $f_{S^*,\varepsilon}$ of $F'(S^*, \varepsilon)$ is indeed a modular function. (proof) From (19), $f_{S^*,\varepsilon}$ is a linear combination of $\{h_i\}_{i=1}^d$. Since an arbitrary linear combination of modular functions is modular as well, it suffices to show $h_i$ is modular. From the definition of $h_i$ (line 245), we can see that all $X, Y \subseteq [n]$ satisfy $h_i(X) + h_i(Y) = h_i(X \cup Y) + h_i(X \cap Y)$. In fact, both sides equal to $-2$ if $X, Y \ni i$ ($\Leftrightarrow X \cap Y \ni i$), to 2 if $X, Y \not\ni i$ ($\Leftrightarrow X \cup Y \not\ni i$), and to 0 otherwise.

> What is the connection between $\hat{f}$ on line 248 and $\hat{f}$ on line 254?

Both have the same expectation given in (19) but follow different distributions. Line 248 gives the definition of $\hat{f} \sim F(S^*, \varepsilon)$ and line 254 is for $\hat{f} \sim F'(S^*, \varepsilon)$. A difference of them is mentioned in lines 255-256. Besides, $\hat{f}(X)$ are independent for different $X$ if $\hat{f} \sim F'(S^*, \varepsilon)$ while $F(S^*, \varepsilon)$ does not have this property.

Dear Reviewer #3:

> Line 172: The range of f is missing;    Thanks for pointing out the typo. We will fix it in the revised version.

> It would be nice if there are experimental results that confirm the effectiveness of the proposed method.

We agree with your suggestion. We consider the empirical evaluation of our method to be an important future work.

[Meta-Review · NeurIPS 2019]

Submodular optimization under stochastic oracle is an important direction and has been studied in the maximization setting. This paper, considers submodular minimization under stochastic oracle access and provides an upper bound on the sample complexity. Based on my own reading, I believe the paper will be of interest to the NeurIPS audience and makes an interesting contribution. However, I urge the authors to include the missing references to stochastic submodular maximization to make the distinction more clear. (i) Gradient Methods for Submodular Optimization (ii) Stochastic Submodular Maximization: The Case of Coverage (iii) Conditional Gradient Method for Stochastic Submodular Maximization: Closing the Gap (iv) Submodular Optimization Under Noise Another point that was brought up during the discussions was the sample complexity lower bound. Again, in the maximization setting, this question has been recently resolved in the following paper "Stochastic Conditional Gradient++". Providing a similar result in the minimization setting would be of great interest. All in all, I would like to recommend this paper to NeurIPS as it adds to our understanding of such fundamental optimization problems.